# ZERO-SHOT GOAL-DIRECTED DIALOGUE VIA RL ON IMAGINED CONVERSATIONS

## ABSTRACT

Large language models (LLMs) have emerged as powerful and general solutions to many natural language tasks. However, many of the most important applications of language generation are interactive, where an agent has to talk to a person to reach a desired outcome. For example, a teacher might try to understand their student's current comprehension level to tailor their instruction accordingly, and a travel agent might ask questions of their customer to understand their preferences in order to recommend activities they might enjoy. LLMs trained with supervised fine-tuning or "single-step" RL, as with standard RLHF, might struggle which tasks that require such goal-directed behavior, since they are not trained to optimize for overall conversational outcomes after multiple turns of interaction. In this work, we explore a new method for adapting LLMs with RL for such *goal-directed dialogue*. Our key insight is that, though LLMs might not effectively solve goal-directed dialogue tasks out of the box, they can provide useful data for solving such tasks by simulating suboptimal but human-like behaviors. Given a textual description of a goal-directed dialogue task, we leverage LLMs to sample diverse synthetic rollouts of hypothetical in-domain human-human interactions. Our algorithm then utilizes this dataset with *offline reinforcement learning* to train an interactive conversational agent that can optimize goal-directed objectives over multiple turns. In effect, the LLM produces examples of possible interactions, and RL then processes these examples to learn to perform more optimal interactions. Empirically, we show that our proposed approach achieves state-of-the-art performance in various goal-directed dialogue tasks that include teaching and preference elicitation.

## 1 INTRODUCTION

Large language models (LLMs) have become very effective at performing a variety of real-world natural language tasks, including open-ended question-answering (Pyatkin et al., 2022), summarization (Paulus et al., 2017; Wu & Hu, 2018; Böhm et al., 2019), code generation (Chen et al., 2021b; Rozière et al., 2023; Zhong & Wang, 2023), and general problem-solving (Wei et al., 2023). While LLMs shine at producing compelling and accurate responses to individual queries, their ability to engage in *goal-directed* conversation remains limited. They can *emulate* the flow of a conversation, but they generally do not aim to accomplish a goal through conversing. For example, we can prompt an LLM to act as a travel agent, and it will produce realistic responses that a human may rate as helpful. But it will not intentionally try to maximize the chance of planning a desirable itinerary for the human. In practice, this manifests as a lack of clarifying questions, lack of goal-directed conversational flow, and generally verbose and non-personalized responses.

The difference between an agent that simply mimics the flow of a conversation and one that pursues conversational goals becomes more apparent when we consider how *uncertainty* influences the conversation. Whether you as the end user are asking the agent to instruct you about a new AI concept, or to plan an itinerary for an upcoming vacation, you have priviledged information which the agent does not know, but which is crucial for the agent to do the task well; e.g., your current background of AI knowledge matters when learning a new concept, and your travel preferences matter when you plan a vacation. A *goal-directed* agent would gather the information it needs to succeed, perhaps by asking clarification questions (e.g., are you an active person?) and proposing partial solutions to get feedback (e.g., how does going to the beach sound?). However, today's LLMs largely fail at this, and are more likely to attempt a helpful but poorly informed guess right away than

to ask appropriate questions. And as we will show in the experiments, even when carefully prompted to gather information, they comply but generate verbose and overwhelming questions that are not good at getting the right information.

In principle, reinforcement learning (RL) can offer a very powerful tool for bridging this gap: LLMs trained with RL to achieve conversational goals (such as maximizing the probability that the user will accept the planned itinerary) could take goal-directed steps, ask clarifying questions, elicit preferences, be very clear and concise in its responses, and maybe even build a rapport with the user. But RL requires data, either in the form of online interactions with a human simulator, or offline human-human interactions. Online data can be computationally difficult obtain, and offline data must be carefully curated to optimize desirable properties such as coverage and diversity (Fu et al., 2020b; Gulcehre et al., 2020; Kumar et al., 2022).

Our key idea is that we can enable *zero-shot goal-directed dialogue agents* by tapping into what LLMs are great at — emulating diverse realistic conversations; and tapping into what RL is great at — optimizing multi-step objectives. We propose to use LLMs to "imagine" a range of possible task-specific dialogues that are often realistic, but where the LLM does not optimally solve the task. In effect, the LLM can imagine what a human *could* do, but not to what an optimal agent *should* do. Conversations are then generated based on sampled hidden states. We train an agent to engage in goal-directed conversation by training offline RL on the resulting dataset.

Our main contribution is a zero-shot RL algorithm that effectively optimizes for goal-directed dialogue tasks. Rather than directly using pretrained LLMs as optimal agents, our method aims to leverage their strength in emulating diverse, human-like, but suboptimal conversations to generate data, which can then be provided to an RL algorithm to actually discover more optimal behaviors. We propose a novel system called the *imagination engine* (IE) that generates a dataset of diverse, task-relevant, and instructive dialogues to be used to train downstream agents. We evaluate our approach on tasks involving teaching of a new concept, persuasion, and preference elicitation. Our experimental results include a user study that compares agents trained with our method to prompted state-of-the-art LLMs, showing that our method can attain significantly better results in interactive conversations even when using models that are orders of magnitude smaller than the prompt-based baseline.

## 2 RELATED WORK

**Language models.** Language models, particularly LLMs, have shown impressive capabilities in text generation (Ghazvininejad et al., 2017; Li et al., 2017; Holtzman et al., 2018; Radford et al., 2019; Yang & Klein, 2021), translation (Gu et al., 2017), question answering (Pyatkin et al., 2022), summarization (Paulus et al., 2017; Wu & Hu, 2018; Böhm et al., 2019), and code generation (Chen et al., 2021b; Zhong & Wang, 2023). However, success at most of these tasks is largely enabled by supervised learning, and does not require reasoning through multiple steps of interaction to optimize a long-term objective. LLMs have been fine-tuned via supervised learning to engage in dialogue with human users to some success (He et al., 2018; Shuster et al., 2022b;a), but primarily to produce realistic responses and not to accomplish an underlying goal.

**RL for language models.** Many existing LLMs leverage reinforcement learning (RL) fine-tuning, where a reward model is learned from feedback directly from human experts (Ziegler et al., 2020; Stiennon et al., 2020; Wu et al., 2021; Nakano et al., 2022; Bai et al., 2022a; Christiano et al., 2023) or secondhand from a handcrafted AI system (Bai et al., 2022b), and is then used to fine-tune the LLM via an RL objective. While finetuning is primarily done via online RL, recent approaches proposed tuning LLMs from offline data (Rafailov et al., 2023; Gulcehre et al., 2023). By doing so, LLMs are able to faithfully follow human instructions, or *prompts*, and can therefore act as general problem solvers by prompt engineering (Ouyang et al., 2022). While effective, one stark downside of RL fine-tuning approaches is that they only consider bandit objectives. Specifically, in RL fine-tuning, LLMs are trained to maximize the learned reward model within a single-step response, and not over the course of a multi-step dialogue. As a result, if the best response to a query is unknown due to latent information, such as intentions or preferences, by the user, traditional LLMs will only provide the best possible "guess" response in one step, and not attempt to gather additional information in order to respond more optimally. Notably, Glaese et al. (2022) propose learning an information-seeking agent, but again consider a single-step objective based on maximizing helpfulness, and do not consider nor evaluate on tasks where gathering information is used to accomplish a long-term goal; the approach also relies on human raters being able to identify useful information-seeking actions.

**Goal-directed dialogue.** There has been numerous prior works on learning models to accomplish tasks via conversations beyond maximizing informativeness or humanness. Goal-directed dialogue, or alternatively task-oriented dialogue, can be formulated as an MDP from which agents can be trained using RL. Online RL methods to optimize dialogue agents typically require a simulator of human behavior, that is usually either handcrafted, or learned as a fixed model (Carta et al., 2023; He et al., 2018; Gašić et al., 2011). Moreover, they involve continual collection of new samples, which incurs a large computational cost in tasks where humans exhibit complex and nuanced behaviors, and are often prone to reward "hacking" (Skalse et al., 2022). Alternatively, offline RL approaches have also been considered that only require a static dataset of dialogues (Jaques et al., 2019; Jang et al., 2022; Verma et al., 2022; Snell et al., 2023). Notably, Verma et al. (2022) propose an offline RL algorithm to solve a goal-directed dialogue based on negotiations using a dataset of conversations between human speakers. However, in order for offline RL to improve over supervised learning, the dataset must be carefully curated to optimize desirable properties such as coverage and diversity (Fu et al., 2020b; Gulcehre et al., 2020; Kumar et al., 2022), which may limit its practicality. Orthogonally, dialogue benchmark datasets have been created that aim to evaluate the capabilities of agents at accomplishing various tasks such as question-answering (Budzianowski et al., 2020), customer service (Chen et al., 2021a), and negotiation (He et al., 2018). However, many such datasets are for tasks that do not necessitate personalizing the agent's responses to each human. In this paper, we consider goal-directed dialogue tasks where humans behave differently due to latent factors, and agents must gather information and personalize to each human. Because of this added complexity, curating a human-human dataset with diverse enough human behaviors can be prohibitively difficult.

**Knowledge distillation.** Our proposed imagination engine can be considered an instance of knowledge distillation (Hinton et al., 2015), where knowledge from a large model (in our case, LLMs) is used to train a smaller model. Recently, this has become popular with LLMs acting as the teacher model, and synthetically generating new training examples for the smaller model (Taori et al., 2023; Chiang et al., 2023; Kim & Rush, 2016). While our approach is similar in principle, all prior approaches consider only downstream supervised learning objectives. To our knowledge, we are the first to do synthetic dialogue generation for RL.

## 3 PRELIMINARIES

**Markov decision processes.** To formulate dialogue as a decision making problem, we use the formalism of the Markov decision process (MDP), given by a tuple $M = (\mathcal{S}, \mathcal{A}, P, r, \rho, \gamma)$, where $\mathcal{S}$ is the state space, $\mathcal{A}$ is the action space, $P$ is the transition function, $r$ is the reward function, $\rho$ is the initial state distribution, and $\gamma$ is the discount factor. When action $a \in \mathcal{A}$ is executed at state $s \in \mathcal{S}$, the next state is sampled $s' \sim P(\cdot|s, a)$, and the agent receives reward $r$ with mean $r(s, a)$.

**Goal-directed dialogues as MDPs.** Goal-directed dialogue can be viewed as an MDP, where states are sequences of tokens from a finite vocabulary $\mathcal{V}$ (Ramamurthy et al., 2023). All tokens that the agent initially observes are used as our initial state, $s_0 = (x_0, \ldots, x_m)$, where $x_i \in \mathcal{V}, \forall i \in [m]$. At timestep $t$, an action $a_t \in \mathcal{V}$ is some token in the vocabulary. As long as $a_t$ is not a special end-of-sequence $\texttt{<EOS>}$ token, the transition function deterministically appends $a_t$ to state $s_t$ to form $s_{t+1}$. Otherwise, the agent observes (potentially stochastic) responses from their conversational partner $o_t = (y_0, \ldots, y_n)$, which also consist of tokens in the vocabulary; then, the transition function appends both $a_t$ and responses $o_t$ to state $s_t$. This continues until the last timestep $T$ where we obtain a state $s_T$ and the agent receives a deterministic reward $r(s_T)$.

In many real-world tasks that require dialogue with a human, humans exhibit a range of different behaviors. For example, in a travel agent task, humans will respond differently to the agent according to their own activity interests, budget, and other personal factors. Such factors are often latent, but affect how an optimal agent should respond. Rather than conventional MDPs, these tasks can instead be formulated as hidden parameter MDPs (Doshi-Velez & Konidaris, 2013), given by a tuple $M = (\mathcal{S}, \mathcal{A}, \mathcal{Z}, P, r, \rho, \gamma)$, where $\mathcal{Z}$ also parameterizes the transition and reward functions. In practice, solutions to hidden parameter MDPs do not need to model $\mathcal{Z}$ explicitly, and instead use a sequence model (i.e., a standard language model) to handle implicitly infer it from the history of observations. Nevertheless, we view $\mathcal{Z}$ as helpful formalism for understanding why information-gathering is important in effective dialogue agents.

**Reinforcement learning.** The goal of reinforcement learning (RL) is to learn a policy $\pi$ that maximizes the expected discounted return $\sum_{t=0}^{\infty} \gamma^t r_t$ in an MDP. The Q-function $Q^\pi(s, a)$ for a

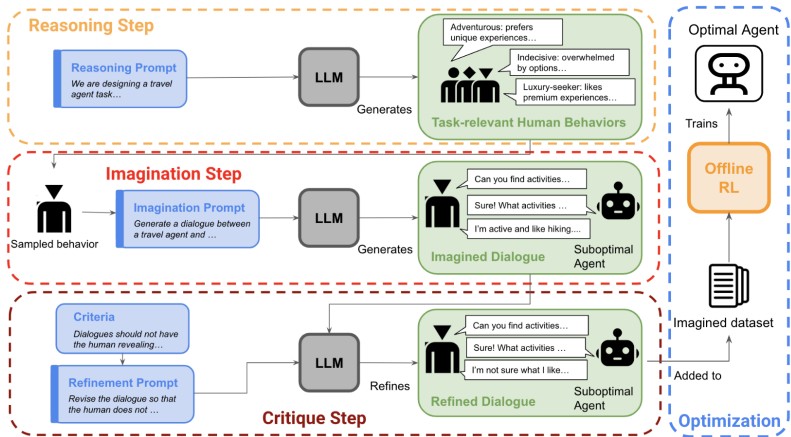

Figure 1: Diagram illustrating our proposed approach, where an imagined dataset of dialogues between humans and a potentially suboptimal agent is synthesized by our imagination engine, then used to train a downstream RL agent. Blue boxes indicate handcrafted quantities.

policy $\pi$ represents the discounted long-term reward attained by executing $a$ given state $s$ and then following policy $\pi$ thereafter. $Q^\pi$ satisfies the Bellman recurrence:

$$Q^\pi(s, a) = \mathbb{B}^\pi Q^\pi(s, a) = r(s, a) + \gamma \mathbb{E}_{s' \sim P(\cdot|s,a), a' \sim \pi(\cdot|s')} [Q(s', a')]$$

The value function $V^\pi$ is the expectation of the Q-function $V^\pi(s) = \mathbb{E}_{a \sim \pi(\cdot|s)} [Q^\pi(s, a)]$. The expected discounted return can be expressed as $J(\pi) = \mathbb{E}_{s_0 \sim \rho} [V^\pi(s_0)]$. In offline RL, we are provided with a dataset $\mathcal{D} = \{(s_i, a_i, s'_i, r_i)\}_{i \in [N]}$ of size $|\mathcal{D}| = N$, generated by an unknown behavior policy $\pi_\beta$ (which might correspond to a mixture of multiple policies). The offline RL setup is particularly useful when online interaction with the real world is costly or unavailable.

## 4 Reinforcement Learning on Imagined Conversations

In this paper, we present an approach for *zero-shot* training of agents in a goal-directed dialogue task. Rather than traditional offline RL approaches that require a curated dataset $\mathcal{D}$ of human-human data, the only input required by our system is a task description $D$. The primary novelty of our proposed approach is an *imagination engine* (IE) that enables the generation of a diverse dataset $\widehat{\mathcal{D}}$ of task-relevant dialogues for any task description $D$. Then, once the dialogue dataset $\widehat{\mathcal{D}}$ is generated, we optimize for an agent $\widehat{\pi}$ via offline RL on the imagined dialogues. We go over both parts in detail.

### 4.1 Imagination Engine: Synthesizing Diverse Task-Relevant Dialogues

We assume access to a LLM $P_{\mathsf{LLM}}(\cdot \mid p)$ that can be used to generate a response for any prompt $p$. The IE consists of three steps, which are outlined in Figure 1 and we describe below. We also provide explicit examples of the process (including prompts used) for two different tasks in Appendix A.

**Reasoning step: synthesize task-relevant personas.** Recall that goal-directed dialogues can be formulated as hidden-parameter MDPs with hidden space $\mathcal{Z}$, where each human has a different $z \in \mathcal{Z}$ that affects how they behave, and how an agent should optimally respond. Without access to prior data, we would naïvely rely on having task-specific knowledge of $\mathcal{Z}$. However, our insight is that LLMs contain a much wider domain of knowledge than any individual human, and therefore can provide task-specific knowledge when humans cannot. Therefore, we propose querying $P_{\mathsf{LLM}}(\cdot \mid f_r(D))$, where $f_r(D)$ is a *reasoning prompt* using task description $D$; the prompt asks the LLM to output textual descriptions of personas $\phi(z)$ for $z \in \mathcal{Z}$. These descriptions $\phi(z)$ can be used to generate human responses under different behaviors (Serapio-García et al., 2023; Park et al., 2023).

**Imagination step: generate synthetic dialogues.** The goal in this step is to imagine dialogues $\tau$ between a (potentially suboptimal) agent and a human. Formally, this involves generating synthetic rollouts in the underlying hidden parameter MDP. Note that in real world, samples from both the transition function $P$ and behavior policy $\pi_\beta$ of the MDP are simply human responses. Therefore, successfully synthesizing trajectories can be reduced to simulating human-human dialogue.

In order to accomplish this, we leverage LLMs to generate synthetic dialogues between an agent and human, where we condition generation on how the human behaves, and the reward that the agent achieves. This is done as follows. First, we sample $\phi(z)$ for some persona $z \in \mathcal{Z}$ that we obtained in the reasoning step, and also sample $r \in \{0, 1\}$ indicating whether the agent fails or succeeds in the generated dialogue. The assumption of binary rewards is only to be task-agnostic, and more expressive rewards can be considered if they exist for a particular task. A conditional dialogue can be sampled $\tau \sim P_{\text{LLM}}(\cdot \mid f_i(D, \phi(z), r))$ where $f_i(D, \phi(z), r)$ is an *imagination prompt* that asks the LLM to output a dialogue between two humans that is task-relevant, and where the human behaves according to $z$ and the agent ultimately achieves reward $r$.

**Critique step: refine dialogues.** Though the synthetic dialogues from the imagination step are mostly plausible interactions, the humans in the dialogue sometimes produce potentially unrealistic responses. For example, humans in the dialogues often reveal their underlying personas, without the agent asking any questions, or even building general rapport with them. Since inferring the human's persona is an important skill we want downstream learning agents to acquire, we want information-gathering strategies to be reflected in the imagined conversations, even if they are not deployed optimally or strategically (as would be required for the optimal agent).

To remedy this, we propose revising the imagined dialogues based on a set of criteria $c$ on what constitutes pedagogical conversations for our downstream learning. Our criteria $c$ are task-specific but generally include the following principles: (1) the "human" should not reveal their latent behavior immediately, but only make it apparent gradually through back-and-forth conversation with the agent; (2) the human's sentiment at the end of the dialogue should accurately reflect the reward that the agent achieves. Our criteria can be used analogously to a constitution to improve the quality of generated responses (Bai et al., 2022b). Formally, we sample a revised dialogue $\tau' \sim P_{\text{LLM}}(\cdot \mid f_c(D, \tau, c))$ from the same LLM where *critique prompt* $f_c(D, \tau, c)$ encapsulates the original dialogue and criteria $c$. Iterating the imagination and critique steps allows us to curate a dataset $\widehat{\mathcal{D}}$ of diverse dialogues.

## 4.2 RL Optimization on the Imagined Dataset

While the imagination engine can produce plausible dialogues, this does not by itself produce effective agents – that is, we use LLMs to synthesize plausible scenarios, including strategies that an agent may take, but not necessarily what an optimal agent should do. In order to determine the optimal strategy that an agent should actually take to achieve a desired outcome, we require multi-step RL to optimize an agent to maximize probability of succeeding at the desired task. Therefore, the main question we aim to answer in this section is the following: *How do we use a static dataset of synthetic dialogues to train an RL agent?* Our solution involves running offline value-based RL to learn a policy purely from the synthetic dataset, without any on-policy samples.

Before running offline RL, we need to postprocess the dataset of synthetic dialogues into RL training examples. Recall that we constructed a dataset $\widehat{\mathcal{D}} = \{(\tau_i, r_i)\}_{i \in [N]}$ of $N$ imagined dialogues, where each dialogue $\tau_i$ a sequence of tokens in vocabulary $\mathcal{V}$ that constitute utterances between a simulated agent and simulated human. For each dialogue $\tau_i$, we isolate all tokens $a$ by the agent, then generate $(s, a, s', r)$ where state $s$ consist of all tokens before $a$, next state $s'$ consist of all tokens before the next token $a'$ by the agent, and $r = r_i$ only if $s' = \tau_i$ is the full dialogue. Using this procedure, we construct a dataset $\widehat{\mathcal{D}}' = \{(s_i, a_i, s'_i, r_i)\}_{i \in [N']}$.

Then, we run value-based RL to learn a policy $\widehat{\pi}$. Specifically, we learn $\widehat{Q}$ and $\widehat{V}$ functions that estimate the optimal $Q$-function and value function, respectively, and then use these functions to extract a policy $\widehat{\pi}$. The functions can be learned using Bellman recurrence:

$$\widehat{Q} = \arg\min_{Q} \mathbb{E}_{(s,a,s',r)\sim\widehat{\mathcal{D}}'} \left[ \left( r + \gamma \widehat{V}(s') - Q(s,a) \right)^2 \right], \widehat{V} = \arg\min_{V} \mathbb{E}_{s\sim\widehat{\mathcal{D}}'} \left[ \left( \max_{a'} \widehat{Q}(s,a') - V(s) \right)^2 \right].$$

When $\widehat{\pi}$ is a language model, we use these functions in combination with the base language model $\widehat{\pi}_{\text{LM}}$ to extract the policy (Snell et al., 2022), via $\widehat{\pi}(a \mid s) \propto \pi_{\text{LM}}(a \mid s)e^{\beta(\widehat{Q}(s,a)-\widehat{V}(s))}$.

If the policy is learned purely from offline data, naïvely training from value-based RL can suffer from distribution shift (Fujimoto et al., 2018; Kumar et al., 2019), which offline RL algorithms remedy by ensuring that the learned $\widehat{Q}, \widehat{V}$ functions are *pessimistic* (Kumar et al., 2020; Kostrikov et al., 2021). Note that our imagination engine is agnostic to the RL algorithm; in our experiments we use Implicit Language Q-Learning (ILQL) (Snell et al., 2022) for its performance on natural language tasks.

## 5 EXPERIMENTS

**Hypotheses.** Our experiments evaluate our proposed zero-shot dialogue agent training procedure on two goal-directed dialogue tasks. The tasks require the agent to perform information gathering in order to personalize their responses to the user, which necessitates goal-directed conversational strategies such as asking clarifying questions, or building rapport with the user to better understand their intentions. We aim to answer the following research questions:

*1. Is leveraging LLMs in our imagination engine to generate synthetic data to train downstream agents preferred over using them naïvely to behave as agents themselves?*

*2. Is offline RL on the imagined data better than simply using imitation learning on the same data?*

The first research question targets our main hypothesis, that LLMs should be leveraged for generating data rather than for solving goal-directed tasks. The second targets whether the specifics of how we train on the imagined data are important. We hypothesize that while in the average case both BC and RL perform similarly, the contrast between RL and BC agents is noticeable in situations that are not well represented in the imagined conversations. In particular, RL agents should be more robust when humans behave in ways that are not represented in any one dialogue in our imagined dataset, but perhaps in concatenations of multiple dialogues. This is because RL agents are exhibited to compose new strategies via a phenomenon called "trajectory stitching" (Fu et al., 2020a; Levine et al., 2020).

**Approaches.** To answer both questions, we consider the following approaches:

**GPT.** This approach prompts GPT-3.5 (OpenAI, 2022), which is a powerful LLM shown in prior work to be able to effectively solve numerous natural language tasks (Ouyang et al., 2022), to directly behave as the agent. The prompt includes both the task description, as well as the insight that the resulting agent needs to gather information about the human user in order to optimally respond to them. This is the traditional usage of LLMs to solve dialogue tasks.

**IE+BC (ablation).** This version of our approach trains an agent on the imagined dataset generated by our proposed imagination engine, but via a behavioral cloning (BC) objective, where the agent straightforwardly mimics the synthetic data. This is equivalent to supervised fine-tuning on the imagined dataset. This is an ablation of our proposed approach.

**IE+FBC (ablation).** Rather than BC on the entire imagined dataset, this method trains the agent using filtered BC instead, which imitates only the successful trajectories in the dataset. This is another ablation of our proposed approach.

**IE+RL.** This is the full version of our approach, which trains the agent using offline RL. Specifically, we use ILQL (Snell et al., 2022) as the offline RL algorithm.

Across methods that use the imagination engine, we use GPT-3.5 (OpenAI, 2022) as the LLM in the imagination engine to generate synthetic data. However, our downstream agents that are trained on imagined data utilize a much smaller decoder-only GPT-2 model (Radford et al., 2019). This is to show that we only need to leverage state-of-the-art LLMs to generate data, as the resulting agents can be much smaller; this makes our approach practical if computational cost is an important consideration, or if modern LLMs are deployed with only inference APIs, which are both hurdles encountered in our empirical evaluations. For each task, we defer implementation details of our imagination engine (including prompts used and results) and RL optimization to Appendix A.

**User study setup.** We test our two hypotheses in a two-part user study with 12 users. To test whether leveraging LLMs for the imagination engine is better than directly using LLMs as agents through prompting, we use a within-subjects design and have users interact with GPT and IE+RL (in a randomized order to avoid ordering effects) across two tasks, which we describe below. We then ask them to rate the agents, for each task, according to four criteria, on a 5-point Likert scale:

**(A)** How well the agent accomplished the task at hand.
**(B)** How realistic and natural the resulting dialogue was.
**(C)** How effective was the agent at asking questions relevant for information-gathering.
**(D)** Overall satisfaction with the agent.

To provide a more fine-grained evaluation that compares RL-based and BC-based IE agents, we specifically analyze challenging scenarios where humans exhibit particularly unusual or difficult personas. Such scenarios exacerbate the shortcomings of BC-based methods, that simply emulate

| GPT Agent: | IE+RL Agent: |
|---|---|
| H: Hi, I've been hearing about something called "behavior cloning" in AI, but I don't really know what it is. Can you explain it to me? | H: Hi there! I've heard about this thing called "behavior cloning" in AI, but I don't really understand what it is. Can you explain it to me? |
| AI: Certainly! Let's start by understanding your background in AI to provide an explanation that matches your level of knowledge. Please answer the following questions: 1. Have you ever studied or worked with artificial intelligence or machine learning concepts before, even at a basic level? 2. Are you familiar with terms like "neural networks," "training data," or "supervised learning"? 3. Are you interested in a high-level overview or a more detailed explanation ... | AI: Of course! I'd be happy to explain behavior cloning. To start, could you tell me if you've ever come across the terms "artificial intelligence" or "machine learning" before? H: Yes, I've heard those terms, but I'm not exactly sure what they mean. AI: No problem at all! Let's take it step by step. Have you ever used a computer or a smartphone? H: Yes, I use my computer and phone all the time. AI: Great! Imagine that a computer ... |

Figure 2: Comparison of dialogues between GPT and IE+RL agents in instruction task. The IE+RL agent exhibits a much more intelligent strategy of asking incremental questions.

the conversational flow in the data rather than optimizing for the task reward. To test whether the RL agent is more robust when the human behaves in ways not reflected in any one dialogue in the dataset, we analyze the generated data, identify unrepresented behaviors (such as ambiguous or unsatisfied users), and emulate them to generate conversations with the IE+BC, IE+FBC, and IE+ILQL agents. We do this because these are behaviors that are less likely to naturally occur through free-flow interactions. We show these conversations to users and ask them to rate the agents as above. We report snippets of evaluated dialogues in the main paper, and defer full dialogues to Appendix B.

**Synthetic study setup.** In addition to the user study on 12 human users, we also conduct a larger scale evaluation in simulation. Due to space, we defer details to Appendix C.

### 5.1 TASK DESCRIPTIONS

We consider two goal-directed dialogue problems based off of the real-world tasks:

**Instruction.** In this task, a human asks an agent to teach them about some concept they are unfamiliar with. Specifically, the human will ask the agent about one of five concepts in RL: "behavior cloning", "policy gradient", "actor-critic", "model-based reinforcement learning" and "offline reinforcement learning". Though this task is similar to general question-answering (Budzianowski et al., 2020), we consider the case where the agent must additionally tailor their instruction to the background knowledge of the human. Hence, in this task, the background of the human constitutes $\mathcal{Z}$ of the hidden-parameter MDP. Example dialogues for this task are shown in Figure 2.

**Preference elicitation.** Here, the agent must build rapport with the human with the objective of uncovering their underlying preferences. We specifically consider a travel agent task alluded to earlier in our paper, where the agent must recommend personalized activities for the human from a fixed set of activities provided in the task description. We have a set of 18 activities grouped into six categories: nature exploration, beach, wellness, food, shopping, and cultural experiences. In contrast to the instruction task, the space of hidden parameters $\mathcal{Z}$ that affect human behavior is much more complicated. Specifically, in addition to uncovering the human's activity preferences, the agent must also figure out and account for additional factors such as personality, budget, or family dynamic. For example, if the human has an indecisive nature, the agent should be wary of overwhelming them with too many options, and rather offer a few choices at a time.

### 5.2 IS IE BETTER THAN PROMPTING?

We first aim to quantitatively and qualitatively address the first research question: is leveraging LLMs as generators of data for RL more effective than directly using them as agents via prompting? For each of the evaluated tasks, we conduct a user study where users interact with the GPT and IE+RL agent three times each, where the agents are anonymous and the order in

| Task | Metric | GPT Agent | IE+RL Agent |
|---|---|---|---|
| Instruction | (A) | $3.4 \pm 0.21$ | $\mathbf{4.3 \pm 0.18}$ |
| | (B) | $2.3 \pm 0.23$ | $\mathbf{3.8 \pm 0.11}$ |
| | (C) | $3.3 \pm 0.33$ | $\mathbf{4.0 \pm 0.13}$ |
| | (D) | $2.4 \pm 0.14$ | $\mathbf{4.2 \pm 0.08}$ |
| Preference Elicitation | (A) | $3.8 \pm 0.21$ | $\mathbf{4.1 \pm 0.11}$ |
| | (B) | $2.9 \pm 0.19$ | $\mathbf{3.7 \pm 0.10}$ |
| | (C) | $2.9 \pm 0.12$ | $\mathbf{4.0 \pm 0.13}$ |
| | (D) | $3.5 \pm 0.16$ | $\mathbf{4.1 \pm 0.11}$ |

Table 1: Mean and standard error of ratings from users interacting with agents across both tasks.

| GPT Agent: | IE+RL Agent: |
|---|---|
| H: I am looking for recommendations for activities to do around here. | H: I am looking for recommendations for activities to do around here. |
| AI: Of course! I'd be delighted to help you find the perfect activities. To get started, could you tell me a bit more about your interests? Are you looking for adventure, culture, relaxation, great food, shopping, nightlife, nature, history, or wellness experiences? | AI: Hello! Welcome to our travel agency. Can you tell me a bit about what you enjoy during your travels? |
| H: I guess I like being outdoors. | H: Yeah, I guess I like being outdoors sometimes. |
| AI: Great! Enjoying the outdoors can be a wonderful experience. We have several options for outdoor activities. If you enjoy a bit of adventure, you might want to consider an adventure hike through our scenic trails, or ... | AI: That's great! Exploring the outdoors can be quite refreshing. Are you more inclined towards peaceful natural spots or do you enjoy exploring cities and their cultural sites? |
|  | H: Um, I'm not sure. Both sound good to me. |
|  | AI: I completely understand. How about I narrow down some options for you? ... |

Figure 3: Comparison of dialogues between GPT and IE+RL agents in preference elicitation task. The IE+RL agent adapts to the user giving vague responses by asking questions with more narrow options.

| Task | Metric | IE+BC Agent | IE+FBC Agent | IE+RL Agent |
|---|---|---|---|---|
| Instruction | (A) | $2.4 \pm 0.18$ | $2.1 \pm 0.12$ | $\mathbf{3.9 \pm 0.10}$ |
|  | (B) | $2.1 \pm 0.21$ | $2.5 \pm 0.10$ | $\mathbf{3.6 \pm 0.26}$ |
|  | (C) | $1.9 \pm 0.08$ | $2.2 \pm 0.23$ | $\mathbf{4.0 \pm 0.19}$ |
|  | (D) | $1.8 \pm 0.16$ | $1.6 \pm 0.13$ | $\mathbf{3.7 \pm 0.11}$ |
| Preference Elicitation | (A) | $1.4 \pm 0.10$ | $1.6 \pm 0.12$ | $\mathbf{3.8 \pm 0.21}$ |
|  | (B) | $2.7 \pm 0.12$ | $2.4 \pm 0.12$ | $\mathbf{3.1 \pm 0.19}$ |
|  | (C) | $2.5 \pm 0.11$ | $2.2 \pm 0.10$ | $\mathbf{2.6 \pm 0.11}$ |
|  | (D) | $1.9 \pm 0.27$ | $1.5 \pm 0.09$ | $\mathbf{3.4 \pm 0.10}$ |

Table 2: Mean and standard error of ratings from users evaluating pre-generated dialogues by agents in both tasks. The RL agent drastically outperforms the BC and FBC agents when interacting with users that are not well-represented in the data.

which the user interacts with them is random. Afterwards, each user reports ratings for metrics (A) to (D). The results are shown in the Table 1. In both tasks, our proposed IE+RL agent outperforms the GPT agent across all metrics, particularly in terms of the naturalness of the resulting dialogue and user satisfaction.

Next, we qualitatively evaluate dialogues between the human user and each agent. In Figure 2, we show a dialogue snippet between an agent in the instruction task and a human who appears to be a layman unfamiliar with AI and RL concepts. In this example, though the GPT agent does make an attempt at information gathering (since it was explicitly instructed to do so in the prompt), it tries to do so in one long survey, which is cumbersome and inconvenient for the human. In contrast, our IE+RL agent gathers information step-by-step by asking natural, easy-to-answer questions, with each question building intelligently on the previous one. The IE+RL agent first asks if the human is familiar with AI, and when the human responds negatively, asks a friendly follow-up question to gauge how familiar they are with technology in general. Next, in Figure 3, we show dialogue snippets between an agent in the preference elicitation task and a human who appears indecisive and gives vague answers. Our IE+LQL agent adapts and offers more narrow questions for the human to answer, whereas the GPT agent continues to asks overly verbose and non-personalized questions. Full dialogues for both examples can be found in Appendix B.

### 5.3 IS OFFLINE RL BETTER THAN BC?

Next, we address the second research question: is training on the imagined data with RL more effective than directly imitating it with supervised learning? Recall that we posit that RL optimization outperforms imitation learning in challenging scenarios where strategies exactly reflected in the data do not suffice. To get such examples, we pose as humans who exhibit potential challenging behaviors and interact with agents. Specifically, in the instruction task, we consider humans who overestimate their understanding of a particular concept. By doing so, an agent's understanding of the human's knowledge background will not align with their true background, resulting in the user not understanding the agent's explanation. Meanwhile, in the preference elicitation task, we consider users who express discontent with the agent's initial recommendation. For each task, we

| IE+FBC Agent: | IE+RL Agent: |
|---|---|
| H: Can you explain it to me what "offline reinforcement learning" is? | H: Can you explain it to me what "offline reinforcement learning" is? |
| AI: Of course! Before I explain offline reinforcement learning, are you familiar with the basic concepts of reinforcement learning? | AI: Of course! Before I explain offline reinforcement learning, are you familiar with the basic concepts of reinforcement learning? |
| H: I'm familiar with reinforcement learning but not offline. | H: I'm familiar with reinforcement learning but not offline. |
| AI: Great! In that case, offline reinforcement learning, also known as batch reinforcement learning, is a specific approach where an agent ... | AI: Great! In that case, offline reinforcement learning, also known as batch reinforcement learning, is a specific approach where an agent ... |
| H: I'm sorry. I didn't understand your explanation. | H: I'm sorry. I didn't understand your explanation. |
| AI: No problem! Reinforcement learning is a type of machine learning where an agent learns to make decisions through trial and error. The agent interacts ... | AI: My apologies. To help me understand your background a bit better, are you aware of the difference between online and offline learning in other domains of machine learning?... |

Figure 4: Comparison of dialogues between IE+FBC and IE+RL agents in instruction task. The IE+RL agent is much more effective at responding to the user being confused.

| IE+FBC Agent: | IE+RL Agent: |
|---|---|
| H: Hi there! I'm in the process of planning a vacation and I'm looking for some ideas on activities to do during my trip. ... | H: Hi there! I'm in the process of planning a vacation and I'm looking for some ideas on activities to do during my trip. ... |
| H: That sounds really tempting, but I'm concerned about the cost. I want to make sure I stick to my budget for the trip. | H: That sounds really tempting, but I'm concerned about the cost. I want to make sure I stick to my budget for the trip. |
| AI: Absolutely, your satisfaction is our priority. I'll continue to tailor my recommendations to your preferences and budget. If you have any... | AI: I understand your concern. If you are looking for budget-friendly shopping options, our location has ... |

Figure 5: Comparison of dialogues between IE+FBC and IE+RL agents in travel agent task. The IE+RL agent responds to negative feedback from the user by revising their suggestions.

generated three dialogue prefixes of challenging situations, then evaluate the capability of the IE+BC, IE+FBC, and IE+RL agents to recover from them. Then, we show such dialogues to each user in the user study, and ask the user to rate the capabilities of each agent for the same metrics (A) to (D). The results are reported in Table 2, where we see clear improvement of the IE+RL agent, especially in asking effective information-gathering questions. In Figure 4, we show corresponding snippets of conversations with the IE+FBC and IE+RL agents in the instruction task. Here, the user expresses confusion with the agent's explanation. The IE+FBC agent decides to paraphrase the prior explanation, whereas the IE+RL agent decides to ask more questions to understand the user's background better. Then, in Figure 5, we show corresponding examples in the preference elicitation task. Here, the user expresses discontent with the agent's expensive recommendation. Only the IE+RL agent decides to offer cheaper alternatives, whereas the IE+FBC agent appears to ignore the user's dissatisfaction. Full dialogues for both examples can be found in Appendix B.

## 6 DISCUSSION

In this paper, we propose an algorithm that achieves *zero-shot* acquisition of goal-directed dialogue agents. The approach leverages a novel imagination engine, which generates a synthetic dialogue dataset that is task-relevant, realistic, and exhibits diverse behaviors. The imagined dataset can then be used to train dialogue agents via offline RL optimization. The key hypothesis that our work demonstrates is that LLMs should not be used directly as goal-directed dialogue agents, but rather as generators for dialogue that can be used for downstream optimization. We show, on a variety of dialogue tasks including teaching and preference elicitation, that our approach is a much more effective usage of LLMs than traditional approaches that prompt LLMs to act directly as agents. Overall, our approach avoids the careful curation of human-human dialogue traditionally used to train dialogue agents via RL. However, we still require human intervention in the form of task-specific prompts. Future work can aim to automate this process further, so that a zero-shot dialogue agent can be trained from any task description.

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
