# OpenReview forum: "Zero-Shot Goal-Directed Dialogue via RL on Imagined Conversations"
_ICLR.cc/2024/Conference — Submitted to ICLR 2024_

### Official Review · Reviewer_yVGF · 2023-10-29

**Soundness:** 3 good
**Presentation:** 2 fair
**Contribution:** 2 fair
**Rating:** 5
**Confidence:** 4

**Summary:**

This paper argues that LLMs trained with SFT or single-step RL might struggle with tasks that require goal-directed behavior. The authors propose a method to use LLMs to generate useful data for solving such tasks by simulating human-like behaviors. The data are used to train a conversational agent for goal-directed tasks with offline RL, in order to improve over the trained conversations. Their results show that the approach achieves better performance than directly prompting LLMs or training the agents with behaviour cloning, in various goal-directed dialogue tasks. However, there are concerns about how the human annotator evaluates the conversations and why the authors did not choose widely-used task-oriented dialogue benchmark datasets like multiwoz and schema-guided-dialogues (SGD).
In addition, although the results show that the learned agents are better than LLMs in information-seeking and generating less overwhelming responses in some specific tasks or domains, I wonder if the smaller agents are able to handle other domains, where the small agents might not have knowledge. Is it still a good alternative to LLMs in this case? Is it still useful?

**Strengths:**

This paper introduces an approach to generate goal-directed conversations with LLMs and then train a smaller agent to improve over these conversations. The human evaluation results show that the learning agents do generate responses that are more helpful in helping the users complete the tasks and generate less overwhelming responses.

**Weaknesses:**

1. Why not utilize the widely used task-oriented dialogue benchmarks, MultiWOZ and SGD? There are works that leverage LLMs for task-oriented dialogue by training a small model to generate dialogue actions (plans) with RL, guiding LLMs for improved responses [1]. Have you considered comparing with them?
2. From the examples comparing GPT-agent and IE+RL agents, GPT's responses didn't seem significantly inferior. How were the responses scored by the evaluators using the four criteria? Was there consensus in their annotations, and what was the level of agreement?
3. LLMs often produce overwhelming responses, but their strength lies in their capability to converse on a wide range of topics due to their inherent knowledge. While training a smaller model for better goal-oriented conversations for some specific domains and topics might seem more beneficial than directly using LLMs, is such a model able to handle out-of-domain tasks and topics? How does it perform when discussing topics outside its training data? If it can not handle out-of-domain topics, is such a model still practical and useful for real-world applications?
4. I feel that the paper complicates the data generation section by unnecessarily introducing numerous reinforcement learning concepts. Why introduce these concepts when it appear to be a simple data generation process, making it challenging to comprehend?

[1] Li, Z., Peng, B., He, P., Galley, M., Gao, J., & Yan, X. (2023). Guiding Large Language Models via Directional Stimulus Prompting. arXiv preprint arXiv:2302.11520.

**Questions:**

See the weakness part.

---

> ### Author Response · Authors · 2023-11-17
> **Response to Reviewer yVGF (1/2)**
>
> Thank you for your review. Your primary concern was a lack of comparison on more objective benchmarks, or to sophisticated prompting methods. We address both of them by providing a new evaluation, and a comparison to a version of the directional prompting approach to our problem setting. We will address each of your concerns in more detail below.
>
> **Task-oriented dialogue benchmarks**
>
> We considered using task-oriented benchmarks such as MultiWOZ. However, upon closer inspection, we found that the tasks themselves did not involve any intelligent information-gathering, and were more similar to question-answering tasks commonly found in NLP benchmarks. Because of this, we did not feel there was a need for agents to exhibit long-term planning behavior, which would be the primary advantage of using multi-step RL.
> In addition, evaluation of agents in these benchmarks would involve computing a ROUGE or BLEU score, which are not quite representative of solving the task itself but how well agents mimic the data. Because of this, we feel that such benchmarks are tailored for supervised-learning agents rather than RL ones.
>
> However, we agree that evaluation via a purely user study is subjective. To remedy this, we provide a larger-scale evaluation on synthetic agents in the preference elicitation task (in Appendix C of our updated paper), where synthetic agents are LLMs prompted with a specific preference (which is an activity they enjoy doing), and we evaluate an agent’s capability to exactly recover this preference. We did find that because such synthetic humans were prompted with an underlying preference, they often prematurely revealed that preference; for example, if the agent proposes an activity that the synthetic human would not enjoy, the synthetic human would simply point out they would prefer their activity more. This means that it was easy to ultimately achieve success against such synthetic humans. Therefore, we additional measure how well each evaluated agent gets the underlying activity in their first try, as well as how verbose they were (in terms of # tokens per utterance). Because our proposed agents are much better at information-gathering, they get the exact activity in the first try in 44% of our evaluations, compared to 18% by a prompted GPT agent; furthermore, our agents do so in 1/3 as many tokens per utterance (43 compared to 118). All our new results are summarized in Table 4.
>
>
> **Examples comparing GPT agent and IE+RL agent**
>
> We believe the IE+RL agent does much more intelligent long-term planning, specifically in how it gathers information about the user’s underlying intent. We present an example comparing the IE+RL and GPT-agent below (more examples can be found in Appendix B of the paper). Here, the GPT-agent tries to explain behavior cloning:
> ```
> H: Hi, I've been hearing about something called "behavior cloning" in AI, but I don't really know what it is. Can you explain it to me?
> AI: Certainly! Let's start by understanding your background in AI to provide an explanation that matches your level of knowledge. Please answer the following questions:
> 1. Have you ever studied or worked with artificial intelligence or machine learning concepts before, even at a basic level?
> 2. Are you familiar with terms like "neural networks," "training data," or "supervised learning"?
> 3. Are you interested in a high-level overview or a more detailed explanation of behavior cloning?
> ```
> The GPT-agent attempts information-gathering because it is prompted to, but does so in a myopic manner by dumping a survey of questions for the human to answer. In contrast, our IE+RL agent employs a “step by step” plan to investigate how much the human already knows about behavior cloning:
> ```
> H: Hi there! I've heard about this thing called "behavior cloning" in AI, but I don't really understand what it is. Can you explain it to me?
> AI: Of course! I'd be happy to explain behavior cloning. To start, could you tell me if you've ever come across the terms "artificial intelligence" or "machine learning" before?
> H: Yes, I've heard those terms, but I'm not exactly sure what they mean.
> AI: No problem at all! Let's take it step by step. Have you ever used a computer or a smartphone?
> H: Yes, I use my computer and phone all the time.
> AI: Great! So, imagine that a computer or a smartphone could learn to do something by watching you do it. That's a bit like what behavior cloning is.
> H: Oh, interesting. Can you explain more about how that works?
> AI: Sure. Behavior cloning is a …
> ```
> We would argue that this is greatly preferred, as users who are unfamiliar with ML concepts would not be able to answer most of the questions provided in the survey by the GPT-agent. This is reflected in the quantitative ratings in Table 1 that the users provide after interacting with both agents. We believe that the level of agreement among users is high, as we also report standard error which in general is much lower than the observed improvement.

---

> > ### Author Response · Authors · 2023-11-17
> > **Response to Reviewer yVGF (2/2)**
> >
> > **Directional-stimulus prompting**
> >
> > This is a great reference that we added to the related work in Section 2. We see two major concerns with this method: (1) the method considers single-step objectives and not goal-directed multi-step dialogue tasks, and (2) it requires access to both human annotations of hints, and online RL training via some simulator. We can remedy (1) by using the learning signal at the end of dialogues rather than after each response (which reduces tractability), and for (2), we use the synthetic humans considered in our new synthetic evaluation (as mentioned above) as simulators of the environment. However, note that when adapted to our problem setting, the directional prompting approach is essentially discrete prompt optimization. Specifically, we fine-tune a GPT2 model to “write a hint” given a task description, using the reward signal from how well a GPT-agent does with the additional hint in its prompt. Note that we do not have access to any human annotations. We found that after training on 100k online trajectories, the hint produced by GPT2 was still nonsensical and did not improve performance
> >
> > **Generalization capabilities of agents**
> >
> > It is true that our current approach only learns task-specific agents. However, we feel that is more a limitation of computational resources than of the generality of our approach. We believe that for future work, one can extend our approach to the multi-task setting by learning a hierarchical policy where the higher-level attempts to identify the task description from user utterances. However, training on many tasks requires compute resources that we do not currently have access to. We view our paper as presenting a prototype approach, and not a complete LLM product replacement akin to GPT.

---

> ### Author Response · Authors · 2023-11-21
> **Let us know if you have further questions**
>
> Thank you again for your review. Let us know if you have any further concerns or clarifications, and we will do our best to address them!

---

### Official Review · Reviewer_Ntkf · 2023-11-01

**Soundness:** 3 good
**Presentation:** 2 fair
**Contribution:** 3 good
**Rating:** 6
**Confidence:** 3

**Summary:**

The study presents a reinforcement learning (RL) approach for training goal-directed dialogue agents on synthetic dialogues produced by large language models (LLMs). Known as the "imagination engine," this method generates training data from simulated talks instead of large-scale human-generated datasets. This technique yields agents who perform better on goal-oriented activities than typical LLMs, indicating a new direction for conversational AI development—one that can comprehend and accomplish difficult tasks with little to no human oversight.

**Strengths:**

1. It shifts the use of LLMs from direct interaction to data generation for optimization by introducing a zero-shot RL algorithm with a "imagination engine" that creatively creates synthetic conversation datasets for training dialogue agents.

2. Compared to traditional approaches, the method optimizes for goal-directed dialogues more effectively since it trains agents on a variety of human-like talks generated by LLMs that are customized for particular dialogue objectives.

3. The usefulness and efficiency of this approach are demonstrated empirically, as agents trained with it outperform state-of-the-art LLMs in interactive tasks.

**Weaknesses:**

A shortcoming of the work is its somewhat dependent use of human-generated prompts, suggesting opportunities for further development in automating zero-shot dialogue agents' training to work without task-specific human input.

**Questions:**

What are the detailed version specifications and hyper-parameter configurations of GPT-3.5 used in the imagination engine, and how do these parameters affect the generated dialogue quality and diversity?

---

> ### Author Response · Authors · 2023-11-17
> **Response to Reviewer Ntkf**
>
> Thank you for your review. We raised a couple concerns that we address below. If there is any additional concrete evidence or information that we can provide to you, we would be happy to do so.
>
> **Human-generated prompts**
>
> That is indeed a limitation of the current approach. However, we would like to point out that prompt-engineering is a large part of many existing approaches that leverage LLMs. We also alleviate much of the burden of prompt-engineering in the reasoning step, by having the LLM provide the task-specific knowledge required to craft prompts. We show the specific prompts we used for each task in Appendix A of our paper, as one can observe that they are quite natural and not overly-engineered. Further, our approach does not seem to be sensitive to the specifics of the prompts used – we will add to Appendix A some qualitative evidence of dialogues generated from different paraphrases of prompts for comparison. This means that our approach takes out much of the heavy-lifting in prompt-engineering and is much more robust than many contemporary usages of LLMs.
>
> **Hyper-parameter configurations**
>
> We use the gpt-3.5-turbo-instruct model that is the same as the widely used ChatGPT interface. We simply use a temperature of 1 and maximum generation length of 512 tokens. We agree that temperature can potentially have some effect on the dialogue diversity, but found that the default of 1 was enough to generate sufficiently diverse dialogues from the same prompt. We have added this information to implementation details in Appendix A of our paper.

---

> ### Author Response · Authors · 2023-11-21
> **Let us know if you have further questions**
>
> Let us know if you have any remaining questions/concerns, as we would be happy to address them!

---

### Official Review · Reviewer_sDqR · 2023-11-02

**Soundness:** 2 fair
**Presentation:** 2 fair
**Contribution:** 2 fair
**Rating:** 5
**Confidence:** 3

**Summary:**

This paper utilizes LLM to simulate sub-optimal but human-like behavior to produce examples of possible interactions. The algorithm uses the data and offline reinforcement learning to train an interactive conversational agent to learn to perform more optimal interactions. Experiments show that the method achieves the most advanced performance in a variety of goal-oriented conversation tasks.
What contributions does it make:
1.The paper propose a zero-shot RL algorithm that effectively optimizes for goal-directed dialogue tasks.
2.The idea of imagination engine (IE) that generates a dataset of diverse, task-relevant, and instructive dialogues makes sense.

**Strengths:**

1.The experimental analysis is detailed and methodical, and the case is clear and intuitive.
2.The idea of using LLM to imitate human behavior is interesting.

**Weaknesses:**

1.Even thought RL can combine parts of behavious seen form behavior policies in the data, it is not convincing that the RL can take all the long-term planing responsibility in the goal-oriented conversation tasks.
2.The novelty of this paper is limited. The proposed method can be regarded as a pipeline of LLM generation and offline RL training.
3.All the evaluation methods are human evaluation, which are highly subjective.
4.More relevant works should be compared in the experiments.

**Questions:**

Need more details about the evaluators in the experiments, such as their education background.

---

> ### Author Response · Authors · 2023-11-17
> **Response to Reviewer sDqR (1/2)**
>
> Thank you for your review. The reviewer raised concerns regarding the results of the paper being unconvincing, particularly in whether our agents actually perform long-term planning. First, we will present why we believe our existing experiments suggest long-term planning capabilities by our agents, and then, we will show results for a new large-scale synthetic evaluation as further evidence. We agree with the reviewer that RL may not solve “all” components of long-term planning, but we believe it is a significant improvement over existing approaches.
>
> First, we argue that the tasks we consider require long-term planning to be done effectively. This is because successful agents require taking information-gathering steps to better understand the user’s intent before proposing a solution.
>
> Quantitatively, our results show that users greatly prefer interacting with our proposed agent than a baseline that prompts existing LLMs (even though these LLMs are much larger), because our agent shows more competence at information-gathering. Specifically, our agent was much better at maintaining back-and-forth dialogue and asking relevant questions, achieving almost 50% higher ratings in those metrics (see Table 1). Qualitatively, we believe that agents learned via our approach explicitly demonstrate long-term planning behavior. Specifically, the dialogue below (and others shown in Appendix B of our paper) shows that our learned agent employs a “step by step” plan to investigate how much the human already knows about behavior cloning, before giving an explanation tailored to the human’s past responses. This capability was not found in any dialogues by agents obtained from directly prompting GPT (examples can again be found in Appendix B).
> ```
> H: Hi there! I've heard about this thing called "behavior cloning" in AI, but I don't really understand what it is. Can you explain it to me?
> AI: Of course! I'd be happy to explain behavior cloning. To start, could you tell me if you've ever come across the terms "artificial intelligence" or "machine learning" before?
> H: Yes, I've heard those terms, but I'm not exactly sure what they mean.
> AI: No problem at all! Let's take it step by step. Have you ever used a computer or a smartphone?
> H: Yes, I use my computer and phone all the time.
> AI: Great! So, imagine that a computer or a smartphone could learn to do something by watching you do it. That's a bit like what behavior cloning is.
> H: Oh, interesting. Can you explain more about how that works?
> AI: Sure. Behavior cloning is a …
> ```
>
> In addition, we add a larger-scale evaluation on synthetic agents in the preference elicitation task (in Appendix C of our updated paper), where synthetic agents are LLMs prompted with a specific preference (which is an activity they enjoy doing), and we evaluate an agent’s capability to exactly recover this preference. We did find that because such synthetic humans were prompted with an underlying preference, they often prematurely revealed that preference; for example, if the agent proposes an activity that the synthetic human would not enjoy, the synthetic human would simply point out they would prefer their activity more. This means that it was easy to ultimately achieve success against such synthetic humans. Therefore, we additionally measure how well each evaluated agent gets the underlying activity in their first try, as well as how verbose they were (in terms of # tokens per utterance). Because our proposed agents are much better at information-gathering, they get the exact activity in the first try in 44% of our evaluations, compared to 18% by a prompted GPT agent; furthermore, our agents do so in 1/3 as many tokens per utterance (43 compared to 118). All our new results are summarized in Table 4.

---

> > ### Author Response · Authors · 2023-11-17
> > **Response to Reviewer sDqR (2/2)**
> >
> > In addition, the reviewer also raised some other concerns that we aim to address below:
> >
> > **Novelty of paper is limited**
> >
> > The use of LLMs to generate synthetic data to train an RL agent is, to the best of our knowledge, novel. While it combines existing ingredients, it’s a novel take on the role of the LLM, and it solves a problem that existing LLMs are worse at solving. Though the approach may be simple in that it can be summarized as “a pipeline of LLM generation and offline RL training,” we propose using LLMs in the “opposite” way as they are traditionally used for – as simulators for RL rather than as agents themselves.  In addition, the LLM generation process required some careful design as to how to reduce the amount of task-specific prompt engineering. For example, we argue it is non-obvious that LLMs can also be leveraged to reason about different task-specific human behaviors, which can then be used to condition the generation process (see Section 4.1 for more details).
> >
> > **Subjective human evaluation**
> >
> > We agree that human evaluations are subjective. To remedy this, we added a larger-scale synthetic evaluation, as described earlier, where we measured objective metrics such as success rate.
> >
> > **More relevant works should be compared**
> >
> > We additionally evaluated a method asked by a different reviewer, where the prompt given to the GPT agent is optimized via discrete prompt optimization [1]. We found that discrete prompt optimization did not improve upon the GPT agent behavior within a tractable amount of training. As mentioned earlier, we are not aware of any RL baselines to compare against that (1) perform multi-turn goal-directed dialogue, and (2) do so without any human-human data curation. If the reviewer has any that come to mind, we would be happy to include them in our evaluations.
> >
> > [1] https://arxiv.org/pdf/2302.11520.pdf

---

> ### Author Response · Authors · 2023-11-21
> **Let us know if you have further questions**
>
> Thanks for reviewing our work. Before discussion ends, we would appreciate it if you let us know what your remaining concerns are with our work. We will do our best to answer them!

---

### Meta-Review · Area_Chair_LKwg · 2023-12-05

**Metareview:**

This paper presents an approach to optimizing goal-directed dialogue agents using a combination of Large Language Models (LLMs) for data generation and offline reinforcement learning (RL). The core concept involves using LLMs to simulate realistic but sub-optimal human conversations, which are then utilized as training data for RL to develop agents capable of better managing goal-directed tasks. This method marks a significant shift from traditional LLM applications, focusing on interactive tasks requiring nuanced understanding and response strategies. While the experimental results show promise, outperforming state-of-the-art models in various goal-directed dialogues, the paper also faces criticisms. Reviewers have pointed out limitations such as over-reliance on human-generated prompts, the absence of comparison with widely used benchmarks like MultiWOZ and SGD, and potential issues with the model's performance in out-of-domain scenarios. Additionally, the paper's complexity in explaining the data generation process and the subjective nature of its evaluation methods have been highlighted as areas for improvement. Despite these concerns, the paper's novel approach and positive empirical outcomes suggest a valuable contribution to the field of conversational AI.

**Justification For Why Not Higher Score:**

Not entirely sure about novelty.

**Justification For Why Not Lower Score:**

NA

---

### Decision · Program_Chairs · 2024-01-16

Reject